# High Expression of the Lysosomal Protease Cathepsin D Confers Better Prognosis in Neuroblastoma Patients by Contrasting EGF-Induced Neuroblastoma Cell Growth

**DOI:** 10.3390/ijms23094782

**Published:** 2022-04-26

**Authors:** Eleonora Secomandi, Amreen Salwa, Chiara Vidoni, Alessandra Ferraresi, Carlo Follo, Ciro Isidoro

**Affiliations:** Laboratory of Molecular Pathology, Department of Health Sciences, Università del Piemonte Orientale “A. Avogadro”, Via Solaroli 17, 28100 Novara, Italy; eleonora.secomandi@uniupo.it (E.S.); salwa.amreen@uniupo.it (A.S.); chiara.vidoni@med.uniupo.it (C.V.); alessandra.ferraresi@med.uniupo.it (A.F.); follocarlo@gmail.com (C.F.)

**Keywords:** cancer, lysosomes, prognosis, cell cycle, EGF, growth factor

## Abstract

Neuroblastoma is a malignant extracranial solid tumor arising from the sympathoadrenal lineage of the neural crest and is often associated with *N-MYC* amplification. Cathepsin D has been associated with chemoresistance in N-MYC-overexpressing neuroblastomas. Increased EGFR expression also has been associated with the aggressive behavior of neuroblastomas. This work aimed to understand the mechanisms linking EGFR stimulation and cathepsin D expression with neuroblastoma progression and prognosis. Gene correlation analysis in pediatric neuroblastoma patients revealed that individuals bearing a high *EGFR* transcript level have a good prognosis only when *CTSD* (the gene coding for the lysosomal protease Cathepsin D, CD) is highly expressed. Low *CTSD* expression was associated with poor clinical outcome. *CTSD* expression was negatively correlated with *CCNB2*, *CCNA2*, *CDK1* and *CDK6* genes involved in cell cycle division. We investigated the biochemical pathways downstream to EGFR stimulation in human SH-SY5Y neuroblastoma cells engineered for overexpressing or silencing of CD expression. Cathepsin D overexpression decreased the proliferative potential of neuroblastoma cells through downregulation of the pro-oncogenic MAPK signaling pathway. EGFR stimulation downregulated cathepsin D expression, thus favoring cell cycle division. Our data suggest that chemotherapeutics that inhibit the EGFR pathway, along with stimulators of cathepsin D synthesis and activity, could benefit neuroblastoma prognosis.

## 1. Introduction

Neuroblastoma (NB) is the most common extracranial solid tumor of childhood, accounting for 15% of cancer-related deaths in children. NB is an embryonal malignancy arising during fetal or early postnatal life from neural crest-derived sympathetic cells. It is commonly found in the adrenal medulla or along the sympathetic chain [1]. The broad spectrum of clinical manifestations ranges from spontaneous regression, maturation into a benign ganglioneuroma or, in the worst cases, into an aggressive and metastatic disease [2]. Despite recent advances in early diagnosis and multimodal therapeutic approaches, current treatments remain elusive and ineffective for many patients with high-risk disease, with a 5-year survival rate of less than 50% [3]. A frequent genetic aberration, occurring in 25% of all NB cases and predicting poor outcome, is *MYCN* amplification [1]. *MYCN* drives oncogenic pathways, and the inhibition of its transcription results in reduced NB cell growth, even in non-*MYCN*-amplified NB cell lines [4]. Overexpression of N-MYC protein stimulates the extracellular release of procathepsin D (proCD) precursor, leading to doxorubicin resistance and increased cancer cell survival [5]. Cathepsin D is a ubiquitous soluble aspartic endopeptidase found in acidic intracellular compartments [6]. CD accomplishes bulk protein degradation and mediates the activation of hormones and their precursors as well as the inactivation of mature growth factors through extensive lysosomal degradation [7]. Accordingly, CD-deficient mice and CD-knockdown zebrafish larvae show severe congenital malformations and premature death [8,9]. The defective intracellular sorting and the escape from lysosomal targeting led to aberrant secretion of cathepsin D precursor [10,11], an event associated with increased tumor size, grading and chemoresistance in a variety of malignancies [12,13].

Epidermal growth factor receptor (EGFR) was found to be overexpressed in NB tumor specimens [14], and its signaling was found to be dysregulated in multi-drug resistant NB cell lines [15]. Current knowledge points to the involvement of cathepsin S and B in the attenuation of EGFR signaling and in receptor degradation [16,17]. Whether cathepsin D plays an active role in EGFR-linked growth and progression of neuroblastoma remains to be elucidated. In the present study, we interrogated datasets from the TCGA database to determine the clinical relevance of *CTSD status* in pediatric neuroblastoma patients highly expressing *EGFR*. We found that patients with high *CTSD* and *EGFR* transcript levels showed a better prognosis and longer overall survival than those with high *EGFR* but low *CTSD*. These preliminary data provided the rationale to speculate that cathepsin D could be involved in EGFR regulation of neuroblastoma growth. We tested this hypothesis in engineered neuroblastoma cells in which cathepsin D was overexpressed or knocked down. We found that in the absence of cathepsin D, EGF increased SH-SY5Y cell growth, and conversely, cathepsin D overexpression attenuated EGF-promoted cell proliferation. Notably, EGF-mediated activation of ERK 1/2 was associated with a downregulation of endogenous CD protein level.

The present work demonstrates, for the first time, a novel antiproliferative role of cathepsin D that may be exploited to improve neuroblastoma management and treatment.

## 2. Results

### 2.1. High CTSD Expression Correlates with Better Prognosis in Pediatric Neuroblastoma Patients

First, we focused on the prognostic value of *CTSD* in pediatric neuroblastoma patients, and we found that the 75% of patients at INSS Stage 4 showing low *CTSD* expression had the worst prognosis. A smaller cohort of individuals with high *CTSD* expression manifested a better outcome (Figure 1A,B). The median overall survival for patients with high *CTSD* expression was 67 months, while the median survival for patients with low *CTSD* expression was 60 months (*p* value = 0.1166, not significantly different). At first diagnosis, most patients present with neuroblastoma expressing low *CTSD*, which is associated with metastatic tumors at Stage 4 (Figure 1C).

### 2.2. High CTSD Expression Increases the Overall Survival of Neuroblastoma Patients Highly Expressing EGFR Transcript

EGFR/HER1, a receptor protein involved in cellular growth and invasiveness, is found frequently overexpressed or aberrantly activated in human neuroblastoma cells, and its inhibition or decreased phosphorylation causes tumor growth suppression and apoptosis in neuroblastomas [14,18,19,20]. Therefore, we extended our studies of *EGFR* expression to evaluate its relationship with *CTSD*. The mRNA expression of *CTSD* and *EGFR* were positively correlated (Figure 2A). This finding was unexpected and somehow counterintuitive, given the above data showing better a prognosis in patients bearing a neuroblastoma expressing *CTSD*.

We grouped the cases with high and low expression, and we analyzed the prognostic value in combinatorial groups of tumors based on the respective levels of mRNA expression of *CTSD* and *EGFR*, as follows: High/High, High/Low and Low/High (Figure 2B). Kaplan–Meier overall survival curves indicated that patients with high *CTSD* mRNA expression, and with either high or low *EGFR* expression, had the better prognosis, while the group bearing low *CTSD* mRNA expression and high *EGFR* exhibited the worst prognosis (Figure 2C). These data suggest that in patients overexpressing *EGFR*, the different clinical outcome is strictly influenced by *CTSD status*.

### 2.3. CTSD Gene Expression Negatively Correlates with Genes Involved in Cell Division

To obtain an insight into the functional role of *CTSD* in pediatric neuroblastoma patients, we performed an *in silico* transcriptomic analysis of the genes correlated to it. We retrieved the RNA-seq data (mRNA expression profile) from the TCGA database (TARGET, 2018) and performed a co-expression analysis to identify the most significant differentially expressed genes (DEGs) that were positively (up-regulated genes in red dots) and negatively (down-regulated genes in blue dots) correlated with *CTSD* in patients’ samples, as represented in the Volcano plot (Figure 3A). We then focused on the genes inversely correlated with high expression of *CTSD*. Notably, the main biological processes regulated by the genes negatively correlated with *CTSD* included mitosis, cell cycle progression, G1/S and G2/S cell cycle transition, nuclear division and DNA replication (Figure 3B). To substantiate this finding, we determined the correlation between the expression of *CTSD* and the genes involved in cell cycle progression. We found that the mRNA expression of *CTSD* was significantly negatively correlated with *CCNB2* (G2/mitotic-specific cyclin-B2), *CCNA2* (cyclin-A2), *CDK1* (cyclin-dependent kinase 1) and *CDK6* (cyclin-dependent kinase 6) (Figure 3C–F).

### 2.4. Generation of Transgenic SH-SY5Y Clones Stably Over-Expressing or Silenced for Cathepsin D

To investigate the role of cathepsin D in determining neuroblastoma growth we sought to genetically manipulate its expression. For this, we generated stable transfectants of human neuroblastoma SH-SY5Y cells in which CD was either overexpressed or silenced. We generated several clones and used the ones with the best desired outcome. Sham-transfected cells, which behaved as untransfected cells, served as controls. We employed the Tet-On gene expression system, in which either the overexpression or the knockdown of CD was switched on in the presence of tetracycline. The pcDNA™4/TO or pENTR™/H1/TO vectors were used, in combination with pcDNA6/TR© plasmid, respectively, for generating CD overexpressing- or downregulated- clones. SH-SY5Y were initially transfected with the pcDNA6/TR© vector. Following selection with 0.4 mg/mL blasticidin, cells stably expressing the Tet repressor were used as the host for the pcDNA™4/TO or pENTR™/H1/TO constructs described below. The vectors are shown in Figure 4. As an internal control for endogenous CD level, the Sham clone was produced by transfecting SH-SY5Y cells with the pENTR™/H1/TO empty vector. In double transfected cells, following tetracycline addition to the culture media, the Tet repressor was switched off, and either the overexpression or the knockdown of CD was enabled.

#### 2.4.1. Construction of pENTR™/H1/TO Plasmids for Cathepsin D Knockdown

For CD knockdown, we constructed three pENTR™/H1/TO plasmids carrying three different shRNA specific for human CD. Two double strand oligonucleotides, encoding for two different shRNA for human CD, were designed using the BLOCK-iT RNAi Designer tool (https://rnaidesigner.thermofisher.com/rnaiexpress/ (accessed on 15 January 2018): shRNA486 (complementary to the bp 486–506 of cathepsin D mRNA) and shRNA879 (complementary to the bp 879–899 of cathepsin D mRNA). As a positive control, a double strand oligonucleotide encoding for a previously validated shRNA (complementary to the bp 1163–1182 of cathepsin D mRNA) [21] was also employed in our study. Stable double transfectants clones (clone 5 pENTR/™H1/TO-shRNA486, clone 4 pENTR™/H1/TO-shRNA879 and clone 1 pENTR™/H1/TO-shRNA Ohri) were then selected with 0.8 mg/mL zeocin. Oligonucleotides sequences are shown in Table 1.

Oligonucleotides were cloned into the pENTR™/H1/TO plasmid following the manufacturer’s protocol (BLOCK-iT Inducible H1 RNAi Entry vector kit, Invitrogen, Waltham, MA, USA). Briefly, DNA oligonucleotides were annealed to generate double strand oligonucleotides and inserted into the pENTR™/H1/TO vector using T4 DNA ligase. The plasmids obtained were subjected to DNA sequencing to check for correct insertion of the oligonucleotides into the vector (shown in Appendix A). Next, the plasmids were transfected in SH-SY5Y, and the efficiency of hCD downregulation by each shRNA was assessed by CD immunoblotting (Figure 4B). The shRNA 486 was the most efficient shRNA in downregulating the aspartic protease. For this reason, pENTR™/H1/TO 486 was employed for the generation of tetracycline-inducible CD knockdown SH-SY5Y cells.

#### 2.4.2. Construction of pcDNA™4/TO Plasmid for Cathepsin D Overexpression

Human CD cDNA [11] was subcloned into the pcDNA4™/TO vector. pcDNA 3.1 Zeo (-) carrying wild type cDNA of human CD was digested with EcoRI and the resulting CD cDNA was inserted into pcDNA4™/TO linearized with EcoRI. The plasmid was then transfected in SH-SY5Y and the efficiency of hCD overexpression was determined by CD immunoblotting (Figure 4A).

Finally, the selected clones used for the subsequent studies were assayed for their level of CD expression. As shown in Figure 4C, the KD-CD and Over-CD clones expressed approximately ten times less and ten times more, respectively, the level of mature (enzymatically active) CD than that expressed in the Sham clone. It should be noted that in the SH-SY5Y knockdown clone, CD was barely detectable (Figure 4C).

### 2.5. Overexpression of Cathepsin D Reduces, While Downregulation of Cathepsin D Enhances, the Proliferative Potential of Transgenic SH-SY5Y Clones

We assessed the proliferative potential of the transfectant clones using a colony-forming assay (as detailed in Materials and Methods section). Cells were seeded at a starting density of 2000 cells/well and allowed to grow for 10 days, with the culture medium renewed every 48 h. Quantification of colony formation (Figure 5A) showed that the clonogenic potential was markedly increased in KD-CD cells (2.3 times higher than Sham and 4.0 times higher than Over CD), and markedly decreased in CD-overexpressing cells (halved compared to Sham). Prompted by this finding, we assayed the cell cycle distribution of the cells in the transfectant clones (Figure 5B). In the KD-CD cell population an increased proportion of cells were found in S phase (7.3%), and even more in G2/M phase (23.58%), compared to that observed in Over CD cell population (where the proportions were 5.48% in S phase and 15.5% in G2/M phase). The latter showed a higher proportion of cells arrested in G0/G1 phase (45.52% versus 36.68% in KD-CD cells and 40.72% in Sham cells). Taken together, these data demonstrate that the level of CD expression impacts on the proliferative ability of neuroblastoma cells, and validate the bioinformatic data showing a strong inverse correlation between the expression of *CTSD* and genes involved in cell cycle progression (Figure 3C–F).

### 2.6. Epidermal Growth Factor Stimulates the Proliferation of KD-CD SH-SY5Y Transgenic Cells While Overexpression of CD Contrasts Its Activity

Bioinformatic analysis showed that overall survival was far better in neuroblastoma patients bearing a tumor expressing high level of CD, irrespective of the level of *EGFR* expression (Figure 2). At this point, we used our CD transgenic clones to test the hypothesis that CD could dampen the proliferative signal downstream to EGFR.

We analyzed the behavior of SH-SY5Y cells in response to EGF at a concentration of 20 ng/mL, which is in the range of cell growth stimulation with no toxic side effects in neuroblastomas [22].

Cell growth assay demonstrated that EGF stimulated the growth of the Sham cultures, and to a much greater extent, also that of the KD-CD clone, evident from 24 h incubation onward, whereas the Over CD clone was relatively insensitive to EGF stimulation (Figure 6A). These data were corroborated by immunofluorescence staining of Ki-67, a proliferative nuclear marker, and of p21 ^Waf/Cip1^, a cyclin-dependent kinase inhibitor that prevents entering the cell cycle (Figure 6B). Consistent with the cell growth data, EGF induced the expression of Ki-67 and decreased the expression of p21. Notably, p21 was basally expressed at a higher level in Over CD cells compared to their Sham and KD-CD counterparts, suggesting an arrest in G1/S transition, and consistent with decreased cell growth.

The cytofluorimetric analysis of the cell cycle was in accordance with above findings (Figure 6C). EGF stimulation in KD-CD resulted in a substantial decrease in the percentage of cells in the G0/G1 phase (−9%), with a corresponding increase in S (+3%) and G2/M phases (+6%), compared to the control condition. The increment was higher in KD-CD than in the other two clones. The fraction of cells in G2/M phase following treatment with EGF was 24.56% in Sham, 27.66% in KD-CD and 23.78% in Over CD. Interestingly, in the EGF-treated Sham culture, we observed that the fraction of cells in S phase (11.28%) was closer to that of untreated KD-CD (11.94%).

### 2.7. EGF Reduces Cathepsin D Protein Level and Increases ERK 1/2 Phosphorylation in SH-SY5Y Neuroblastoma Cells

The data above suggested that EGF can modulate cell proliferation in the Sham and KD-CD clones, and to a much lesser extent, in the Over CD clone. We suspected that such a differential effect was related to the ability of EGF to modulate endogenous CD, whose expression was shown to correlate with cell cycle genes. Therefore, we determined the duplication time of the Sham and KD-CD clones with or without EGF stimulation. EGF reduced the doubling time of Sham cells by approximately 23% (from 39 h to 30 h), and of KD-CD cells by approximately 30% (from 27 h to 19 h) (Table 2). Intriguingly, supplementation of EGF to the Sham culture made these cells proliferate, with a doubling time (29.9 ± 1.9 h) close to that of untreated KD-CD cells (26.8 ± 1.2 h). This prompted us to hypothesize that EGFR stimulation could result in CD downregulation. To test this hypothesis, we measured the CD protein content in EGF-treated Sham at different time points. The western blotting data demonstrate that CD was in fact downregulated in neuroblastoma cells challenged with EGF (Figure 7A).

Finally, we assayed the activation of the cell proliferation signaling pathway downstream to EGFR. In particular, we focused on the ERK pathway as this is the main mitogenic signaling triggered by EGF [23]. As confirmation, an increased phosphorylation of ERK 1/2 was observed in cells exposed to EGF (Figure 7B).

## 3. Discussion

CD is an aspartic protease resident in acidic compartments, where it accomplishes the degradation of extracellular proteins internalized by endocytosis or phagocytosis, as well as that of intracellular proteins delivered to lysosomes by autophagy [24]. This function is associated with protein homeostasis and, consequently, cell growth control [10,25]. Under cytotoxic and stressful conditions, cytoplasmic relocation of mature CD from the lysosomes can drive apoptosis [26,27,28,29,30,31,32]. On the other hand, overexpression of the *CTSD* gene, along with defective segregation in the acidic compartments, leads to abnormal secretion of the precursor proCD, which can be found in the culture media and body fluid of tumor bearers [13,33,34,35,36]. In contrast, hypersecretion of proCD elicits oncogenic activities. Extracellular proCD acts as an autocrine and paracrine growth factor for an unknown receptor, triggering RAS/MAPK and PI3K/AKT pathway activation in fibroblasts and human endothelial cells [37,38]. A well-organized vascular network supports tumor growth and favor metastasis. proCD may undergo autoactivation in the acidic tumor microenvironment, thus favoring cancer cell invasion and neo-angiogenesis through the degradation of the extracellular matrix (ECM), liberating growth factors [13,39]. In this context, secreted CD exerts a pro-angiogenic activity by processing the precursors of VEGF-C and VEGF-D, releasing active growth factors [40]. Therefore, it is fundamental that CD is correctly segregated to perform its function in the endosomal–lysosomal compartments, to avoid its aberrant secretion. In neuroblastoma cells, secreted proCD exerted an anti-apoptotic activity, promoted cell survival and contributed to doxorubicin resistance [5]. Here, we report that patients bearing a neuroblastoma that expresses a high level of the lysosomal protease CD benefit from a better prognosis. Further, *in silico* transcriptome analysis revealed that neuroblastoma patients with high *EGFR* and high *CTSD* levels had a better prognosis compared to patients bearing high *EGFR* and low *CTSD*. Accordingly, two-thirds of patients at INSS Stage 4 present with a low level of *CTSD* expression, indicating that neuroblastomas with CD deficiency grow and progress faster than neuroblastomas that express a high level of CD. Therefore, we argued that in neuroblastomas highly expressing CD, the protease is retained intracellularly and can control protein homeostasis and, consequently, the cell cycle. Consistent with this interpretation, overexpression of the *CTSD* gene negatively correlated with a subset of genes associated with cell cycle and proliferation, including *CCNB2*, *CCNA2*, *CDK1*, and *CDK6*.

We validated our hypothesis using transgenic human SH-SY5Y neuroblastoma cells in which CD was either stably overexpressed (under the CMV promoter) or knocked down (by specific short-hairpin RNA). The addition of exogenous EGF leads to receptor activation and enhances NB cell proliferation [20]. SH-SY5Y overexpressing CD showed a higher expression of p21 and were less responsive to EGF stimulation, whereas SH-SY5Y knocked down for CD were basally more proliferative and more responsive to EGF stimulation. Notably, Sham-transfected SH-SY5Y cells, which retain the ability to modify protein level, responded to EGF challenge by increasing cell proliferation, along with downregulating the level of endogenous CD. To our knowledge this is the first report showing such an effect of EGF on CD expression in cancer cells.

The EGF family of growth factors are potent inducers of angiogenesis in vitro and in vivo, and EGFR ligands are frequently released in the tumor microenvironment from cancer and non-cancer cells [41]. The existing crosstalk between these cells is crucial for sustaining tumor growth and for promoting angiogenesis. In fact, the heparin-binding EGF-like growth factor (HB-EGF), when released in the tumor microenvironment, induces endothelial cell proliferation in solid cancers and in multiple myeloma [42,43,44]. The pharmacological inhibition of HB-EGF-EGFR signaling with erlotinib results in anti-angiogenic effects, inhibits cancer cell growth both in vitro and in vivo, and prevents multiple myeloma progression [44].

The inhibition of growth factor receptors is an attractive approach for treating cancers. However, while the HER1-specific tyrosine kinase inhibitor ZD1839 (Iressa, gefitinib) markedly reduces receptor activation and PI3K/AKT signaling, it is not effective on MAPK in neuroblastoma [20]. In addition, alterations of EGFR itself (polymorphisms, variants), overexpression of HER family ligands, and lncRNAs are associated with monoclonal antibody resistance and tumor relapse [45,46,47,48]. Developing insight into the molecular mechanisms of downstream growth factor receptors may help the identification of novel key targets and the development of effective therapeutics.

In this context, it is of relevance that ERK 1/2 activation by EGFR was reduced in the neuroblastoma cells overexpressing CD. Thus, our data may have translational application for the personalization of therapy for neuroblastomas in patients stratified for the expression of *N-MYC*, *EGFR* and *CTSD.* These patients could in fact benefit from a therapy combining an inhibitor of the EGFR signal with a drug inducing the expression and/or stimulating the activity of CD, such as, for instance, the nutraceutical resveratrol [29]. Worthy of note, the latter modulates autophagy [49], interrupts the metabolic crosstalk between cancer and stromal cells [50,51], and suppresses neo-angiogenesis [52,53]. Notably, the anticancer effectiveness of RV, with no toxic side effects, has been reported in several ongoing clinical trials (recorded on clinicaltrial.gov, accessed on 21 April 2022) [54].

Our findings encourage in vitro testing of the effects of resveratrol as adjuvant therapy for neuroblastomas low-expressing cathepsin D and not responding to the EGFR inhibitor gefitinib [55].

## 4. Materials and Methods

### 4.1. Cell Culture and Treatment

Human neuroblastoma SH-SY5Y cells were obtained from the American Type Culture Collection (cod. CRL-2266, ATCC, Rockville, MD, USA). SH-SY5Y cells were maintained under standard conditions (37 °C, 95 *v*/*v*% air: 5 *v*/*v*% CO_2_) in 50% Minimum Essential Medium (MEM, cod. M2279, Sigma-Aldrich Corp., St. Louis, MO, USA) and 50% Ham’s F12 Nutrient Mixture (HAM, cod. N4888, Sigma-Aldrich Corp.), containing 10% heat-inactivated Fetal Bovine Serum (FBS, cod. ECS0180L; Euroclone S.p.A., Milan, Italy), supplemented with 1% Glutamine (cod. G7513, Sigma-Aldrich Corp.) and 1% *w/v* of Penicillin/Streptomycin (cod. P0781, Sigma-Aldrich Corp.). Cultured cells were treated, where specifically indicated, with 20 ng/mL Epidermal Growth Factor (EGF, cod. E5036; Sigma-Aldrich Corp.) dissolved in 10 mM acetic acid. SH-SY5Y stable transfectant clones (Sham, knockdown CD (KD-CD) and Over CD), representing different CD protein levels, were engineered in our laboratory. Plasmids and reagents employed for clone generation were purchased from Invitrogen, Waltham, MA, USA.

### 4.2. Cell Counting, Doubling Time and Cell Cycle Analysis

Cells were plated into 12-well plates (50,000 cells/cm^2^), allowed to adhere for 24 h, and then treated with 20 ng/mL EGF where appropriately indicated. At each time point, cells were trypsinized, collected, and then the cell suspensions were diluted 1:1 with Trypan Blue solution for counting. Time zero refers to the start of treatment, 24 h after plating. Medium was refreshed every day. Cell counting was performed in triplicate for each experimental condition. Doubling time (Dt) was calculated using the free software Doubling Time Online Calculator (http://www.doubling-time.com/compute.php (accessed on 15 July 2021). Cells were fixed in 70% ice-cold ethanol and stored at −20 °C till the start of cytofluorometric analysis. Cells were incubated with RNAse for 30 min at 37 °C, and the DNA was subsequently stained with propidium iodide (PI, 50 μg/mL; cod. P4170, Sigma Aldrich). The stained cells were then analyzed by using a FacScan flow cytometer (FACSCalibur, Becton, Dickinson, Eysins, Switzerland). For each sample, a fraction of 5000 events was assessed. Cytofluorimetric data were elaborated through Flowing software (v2.5.1).

### 4.3. Clonogenic Assay

For clonogenic assay, cells were seeded into 6-well (MW6) plates at a density of 2000 cells/well, and treated with 20 ng/mL EGF. The cells were cultivated for 10 days to allow colony formation [56]. At the end of the experiment, the medium was removed, cells were washed with 1X PBS, and then fixed with methanol for 20 min at room temperature. Next, another wash with 1X PBS was performed, and subsequently colonies were stained with 0.5% crystal violet solution for 30 min. Finally, MW6 plates were washed with distilled water (until the background became clear) and dried at room temperature. Each well was photographed, and the number of colonies formed was estimated by photometric measurements and CellCounter software (v0.2.1.). 

### 4.4. Antibodies

The following primary antibodies were employed for western blotting: mouse anti-β-tubulin (1:1000, cod. T5201; Sigma-Aldrich Corp.), mouse anti-β-actin (1:2000, cod. A5441; Sigma-Aldrich Corp.), rabbit anti-GAPDH (1:1000, cod. G9545; Sigma-Aldrich Corp.), mouse anti-cathepsin D (1:100, cod. IM03; Calbiochem, St. Louis, MO, USA), rabbit anti phospho-ERK 1/2 (Thr202/Tyr204, Thr185/Tyr187) (1:500, cod. 05-797R; Millipore, Burlington, MA, USA) and mouse anti-ERK1/2 (1:500, cod. 05-1152; Millipore). Secondary antibodies employed for immunoblotting were purchased as follows: Horse Radish Peroxidase-conjugated goat anti-mouse IgG (1:10,000, cod. 170–6516; Bio-Rad, Hercules, CA, USA) and Horse Radish Peroxidase-conjugated goat anti-rabbit IgG (1: 10,000, cod. 170–6515: Bio-Rad, Hercules, CA, USA). The following primary antibodies were employed for immunofluorescence: mouse anti-p21 (1:100, cod. sc-817; Santa Cruz Biotechnology, Dallas, TX, USA) and rabbit anti-Ki-67 (1:100, cod. HPA001164; Sigma-Aldrich). Secondary antibodies used for immunofluorescence were purchased as follows: goat-Anti Rabbit IgG Alexa FluorTM Plus 488 (1:1000, cod. A32731; Invitrogen) and Goat-Anti Mouse IgG Alexa FluorTM Plus 555 (1:1000, cod. A32727; Invitrogen).

### 4.5. Western Blotting

SH-SY5Y Sham, KD-CD and Over CD clones were plated at a density of 50,000 cells/cm^2^ on sterile P35 Petri dishes and allowed to adhere. Cells were harvested in RIPA Buffer (0.5% Deoxycholate, 1% NP-40, 0.1% Sodium Dodecyl Sulfate in PBS solution) supplemented with protease inhibitor cocktail and phosphatase inhibitors (0.5 M sodium fluoride NaF and 0.2 M sodium orthovanadate Na_3_VO_4_), and homogenized using an ultrasonic cell disruptor XL (Misonix, Farmingdale, NY, USA). All reagents were supplied by Sigma-Aldrich Corp. Protein content concentration was determined using a Bradford assay and samples were denatured with 5X Leammli sample buffer at 95 °C for 10 min. Equal amounts of protein (30 μg of total cell homogenates) were separated by SDS-PAGE and transferred onto a PVDF membrane (cod.162-0177; BioRad, Hercules, CA, USA). Filters were blocked with 5% non-fat dry milk (cod. sc-2325; Santa Cruz Biotechnology) solution containing 0.2% Tween-20 for 1 h at room temperature (RT). Subsequently, membranes were incubated with specific primary antibodies overnight at 4 °C, followed by incubation with secondary HRP-conjugated antibodies (goat anti-mouse (cod. 170-6516) and goat anti-rabbit (cod. 170-6515)) for 1 h at room temperature. The bands were detected using Enhanced Chemiluminescence reagents (ECL, cod. NEL105001EA; Perkin Elmer, Waltham, MA, USA) and developed with a ChemiDoc XRS instrument (BioRad, Hercules, CA, USA). Intensity of the bands was estimated by densitometry using Quantity One Software (BioRad, Hercules, CA, USA).

### 4.6. Immunofluorescence

SH-SY5Y cell clones were seeded onto sterile coverslips at a density of 40,000 cells/cm^2^, and allowed to adhere and grow before treatment. At the end of the experiment, the coverslips were fixed in ice-cold methanol, permeabilized with 0.2% Triton-PBS, and then re-fixed with methanol. After washing with 1X PBS, coverslips were incubated overnight at 4 °C with specific primary antibodies dissolved in 0.1% Triton PBS + 10% FBS. The following day, the coverslips were washed three times with 0.1% Triton-PBS and incubated for 1 h at room temperature with Goat-Anti Rabbit IgG Alexa FluorTM Plus 488 or Goat-Anti Mouse IgG Alexa FluorTM Plus 555 secondary antibodies, as appropriate. Nuclei were stained with the UV fluorescent dye DAPI (4′,6-diamidino-2-phenylindole). Secondary antibodies and DAPI were dissolved in 0.1% Triton-PBS + 10% FBS. Thereafter, coverslips were mounted onto glasses using SlowFade antifade reagent (cod. S36936; Life Technologies, Paisley, UK) and data acquired by fluorescence microscopy (Leica DMI6000, Leica Microsystems, Wetzlar, Germany). For each experimental condition, different microscopic fields were randomly selected.

### 4.7. Bioinformatic Analysis

Kaplan–Meier curves, correlation studies and biological processes were obtained by extracting clinical data from the TCGA database (www.portal.gdc.cancer.gov/, last accessed on 13 December 2021). RNA-seq and corresponding clinical data (including overall survival *status*, INSS stage and mRNA expression of 20,040 group of genes) of pediatric neuroblastoma patients (TARGET 2018, comprising 248 patients after filtering out datasets with insufficient survival information) were downloaded from the cBioportal.org [57]. Patients were grouped based on the level of mRNA expression; low versus high groups were defined relative to the median expression level of the overall patient cohort. The correlation between the mRNA expression of the relevant biomarker *CTSD,* and the INSS Stage is represented in histograms. Pearson’s and Spearman’s correlation analyses were performed to identify the genes correlated with *CTSD*.

TBtools (https://github.com/CJ-Chen/TBtools/ (accessed on 4 December 2021)) was used to identify differentially expressed genes (DEGs) in correlation with *CTSD*, represented as a Volcano plot. To identify the DEGs, the cut-off criteria were set based on Spearman’s correlation values (i.e., correlation coefficient value greater than +0.45 (positively correlated) or lower than −0.45 (negatively correlated) and *p*-value < 0.0001 (−log10 (*p*-value) threshold was fixed above 5.0)).

DAVID bioinformatics functional annotation tool (https://david.ncifcrf.gov/summary.jsp (accessed on 20 December 2021) was used to analyze Gene Ontology (GO) biological processes and Kyoto Encyclopedia of Genes and Genomes (KEGG) pathways were obtained with the help of negatively-differentially expressed genes. Data are presented in bar graphs displaying the number of transcripts belonging to negatively associated biological processes.

Scatter plots were employed to represent the correlation between the expression of relevant biomarkers in the patient cohort. Pearson’s correlation analyses were performed to identify the correlation between *CTSD* and *CCNB2*, *CCNA2*, *CDK1* and *CDK6* genes. Regression was estimated by calculating Pearson’s correlation coefficients (r) and the relative *p*-values.

*CTSD* and *EGFR* were grouped based on the level of mRNA expression in the neuroblastoma patients. The correlation of *CTSD* and *EGFR* was determined after sub-classification of mRNA expression based on the level of Z-score values, as high (all positive z-score values) and low (all negative z-score values), respectively. Low versus high mRNA expression was defined relative to the median expression level of all patients in the form of a box plot, and used to investigate the relationship between dichotomized *CTSD* and *EGFR* expression. To reduce potential bias from dichotomization, the mRNA expression of *CTSD* and *EGFR* were compared using a *t*-test (Welch Two Sample *t*-test) by R. All cut-off values were set before the analysis, and all the tests were two-tailed.

All statistical analyses were performed using R (3.6.1 version, The R Foundation for Statistical Computing, Vienna, Austria) and SAS software (9.4. version, SAS Institute Inc., Cary, NC, USA). The log-rank test was used to determine statistical significance. *p*-value ≤ 0.05 was considered to be significant. Survival analysis was performed using SAS for the following: *CTSD* expression and mRNA expression level-based groups of *CTSD* and *EGFR*. Survival curves of these two groups were estimated using Kaplan–Meier plots and compared using the Cox regression model, assuming an ordered trend for the three groups as described previously. The log-rank test was used to determine the statistical significance. *p*-value < 0.05 was considered significant.

### 4.8. Statistical Analysis

Statistical analysis was performed with GraphPad Prism 6.0 software (San Diego, CA, USA). Bonferroni’s multiple comparison test after one-way ANOVA analysis (unpaired, two-tailed) was employed. Significance was considered as follows: **** *p* < 0.0001; *** *p* < 0.001; ** *p* < 0.01; * *p* < 0.05. Data are reported as average ± S.D. Unpaired *t*-test analysis was also employed.

## Figures and Tables

**Figure 1 ijms-23-04782-f001:**
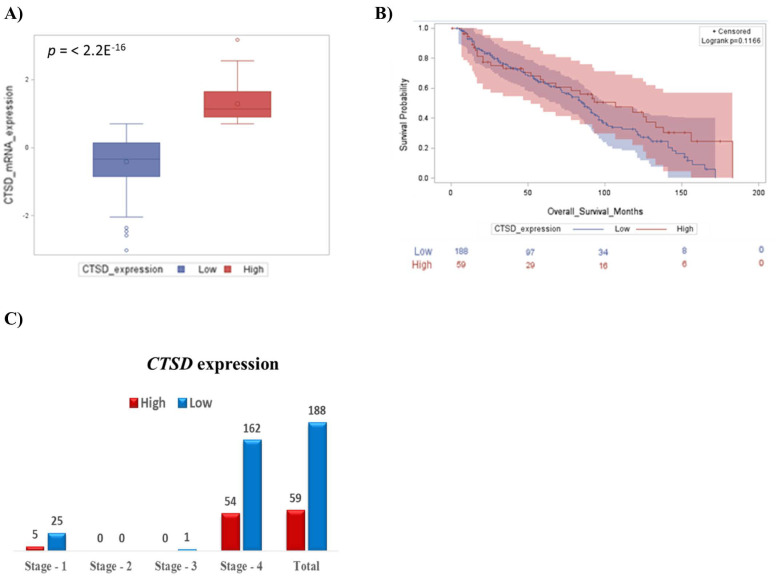
Neuroblastoma patients bearing high *CTSD* transcript level show a better prognosis. (**A**) Box-plot showing the distribution of *CTSD* expression. (**B**) Kaplan–Meier plot representing the overall survival of neuroblastoma patients according to *CTSD* expression levels (high and low). (**C**) Low expression of *CTSD* correlates with INSS Stage 4 in neuroblastoma patients.

**Figure 2 ijms-23-04782-f002:**
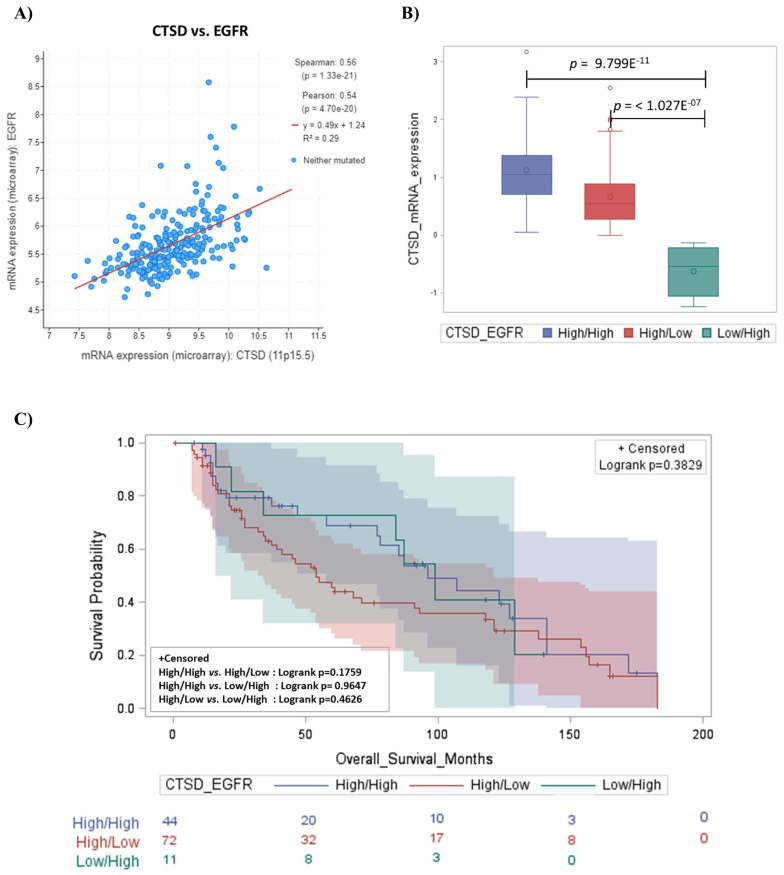
*CTSD* positively correlates with *EGFR* expression and patients exhibit a better prognosis. (**A**) Scatter plot showing the positive correlation between *CTSD* and *EGFR* expression. (**B**) Box-plot representing the distribution of *CTSD* mRNA expression level in different combinations of *CTSD* and *EGFR*—high/high, high/low and low/high groups, respectively. (**C**) Kaplan–Meier plot representing the overall survival of neuroblastoma patients according to combination of *CTSD* and *EGFR*—high/high, high/low, and low/high groups, respectively. Log-rank *p* value for each combination is reported in the box.

**Figure 3 ijms-23-04782-f003:**
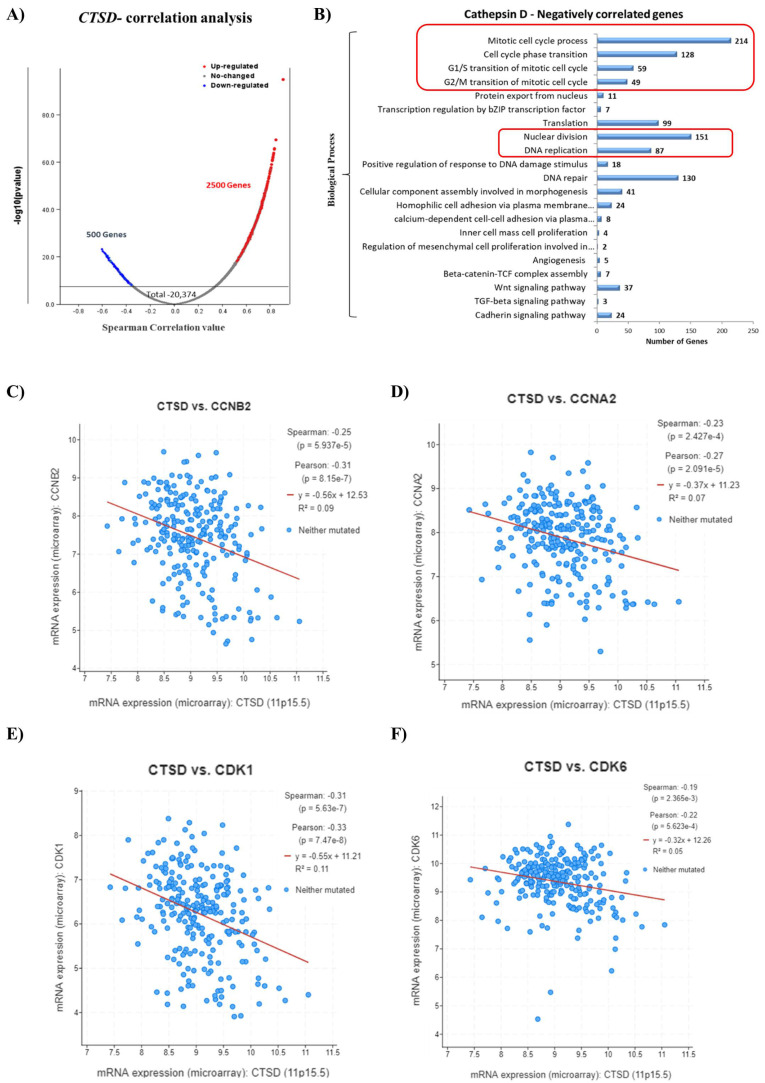
High *CTSD* expression inversely correlates with genes involved in cell cycle progression. (**A**) Volcano plot displaying the differential expressed genes (DEGs). Red dots represent *CTSD*-positively correlated genes, while blue dots represent *CTSD*-negatively correlated genes. (**B**) Graph reporting the negatively correlated biological processes with *CTSD*. (**C**–**F**) Scatter plots showing the negative correlation between *CTSD* and *CCNB2*, *CCNA2*, *CDK1*, *CDK6*, respectively.

**Figure 4 ijms-23-04782-f004:**
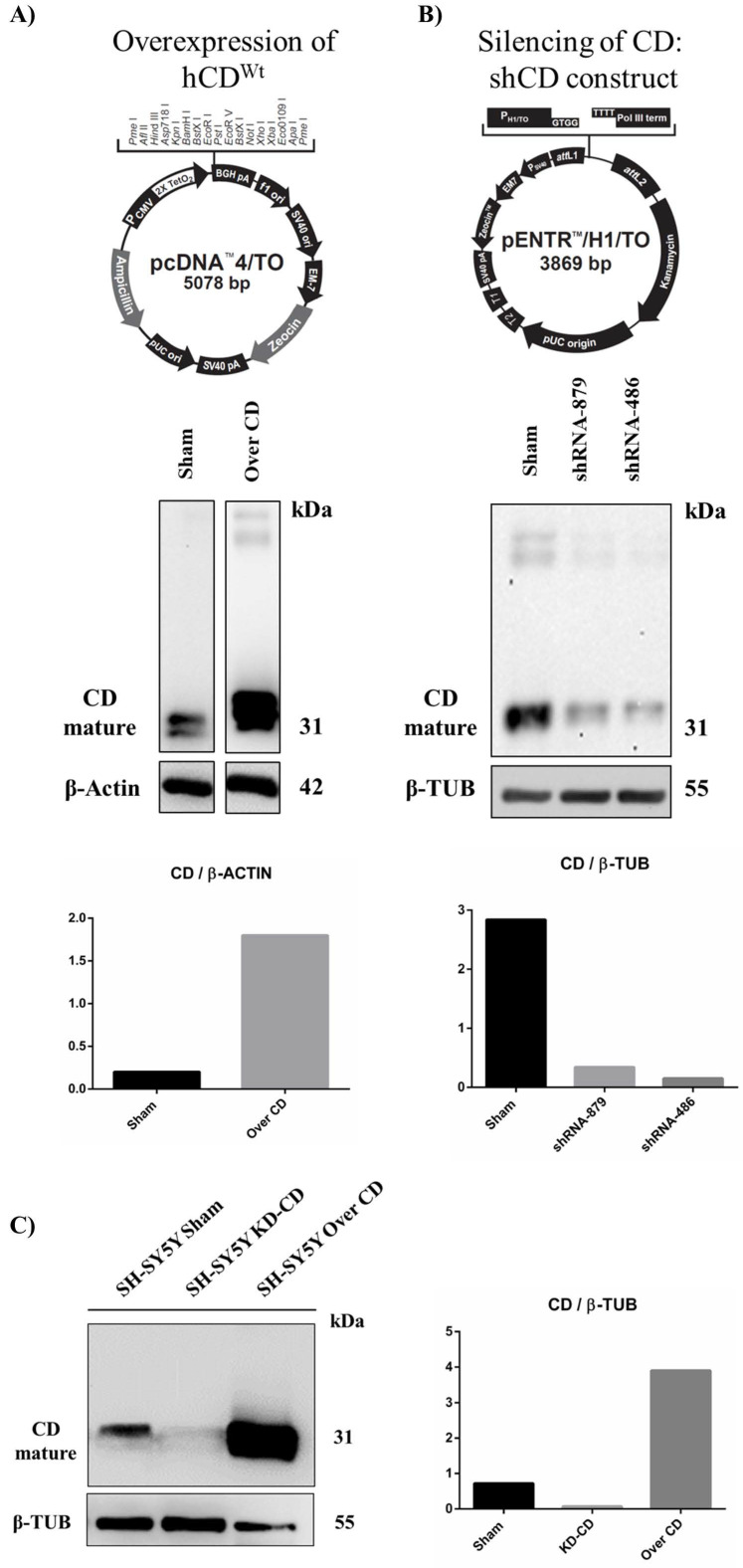
Generation and validation of transgenic SH-SY5Y clones. pcDNA™4/TO (**A**) and pENTR™/H1/TO (**B**) vectors used for cathepsin D overexpression and silencing, respectively. Below, western blot analysis of CD expression is shown. (**C**) Western blot of cathepsin D in SH-SH5Y Sham, 486 (KD-CD) and Over CD clones. Densitometry of the bands is reported in the histogram.

**Figure 5 ijms-23-04782-f005:**
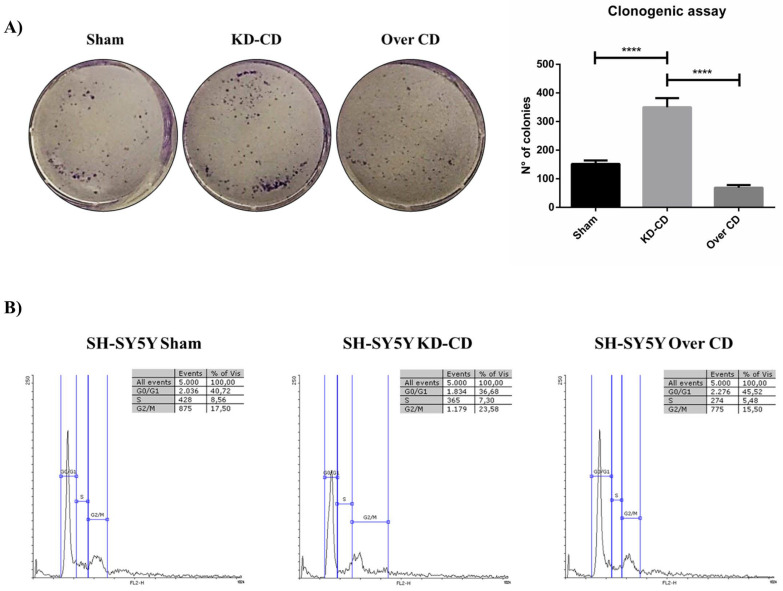
SH-SY5Y Sham, KD-CD and Over CD clones show different growth rates depending on CD expression levels. (**A**) Clonogenic assay and representative graph of the new colonies formed during 10 days of culture. Cells were seeded in 6-well plates and stained with 0.5% crystal violet solution. Images were acquired and colony counting was performed using CellCounter software. Cell growth and number of colonies were estimated through photometric measurements using CellCounter software and are shown in the graph. Data ± S.D. are representative of three independent replicates. Significance was considered as follows: **** *p* < 0.0001. (**B**) Cell cycle analysis performed at 72 h. The percentage of cell populations in different cell cycle phases is reported. Quantification was performed using Flowing software 2.0.

**Figure 6 ijms-23-04782-f006:**
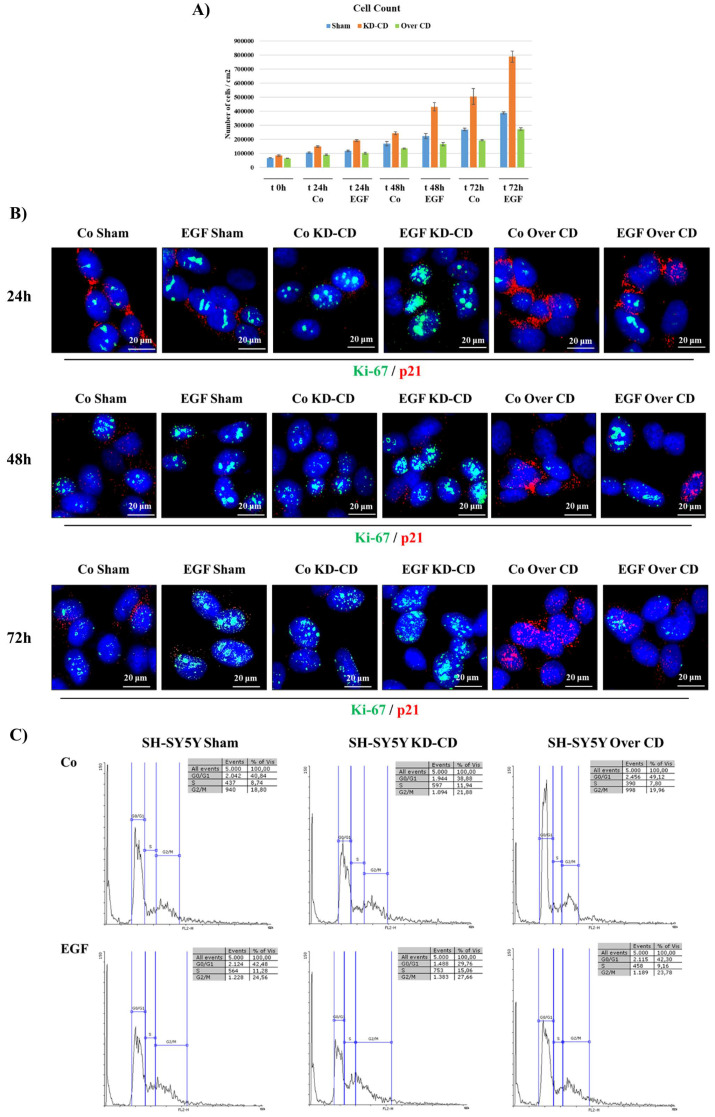
SH-SY5Y KD-CD cells are more sensitive to EGF and show a faster growth compared to CD-overexpressing cells. Assessment of cell proliferation following 20 ng/mL EGF treatment. (**A**) The figure shows a graphical representation of cell count, performed in triplicate for each experimental condition. The treatment was repeated every 24 h, until the end point of 72 h. Time zero is referring to the first day of treatment. (**B**) Immunofluorescence double staining at 24, 48, 72 h. Cells were stained for Ki-67 (green)/p21 (red). Scale bar = 20 μm; magnification = 63X. Representative images of different fields for each experimental condition are shown. (**C**) Cell cycle analysis performed on SH-SY5Y clones after 72 h of EGF. The percentage of cell populations in different cell cycle phases is reported. Quantification was performed by Flowing software 2.0.

**Figure 7 ijms-23-04782-f007:**
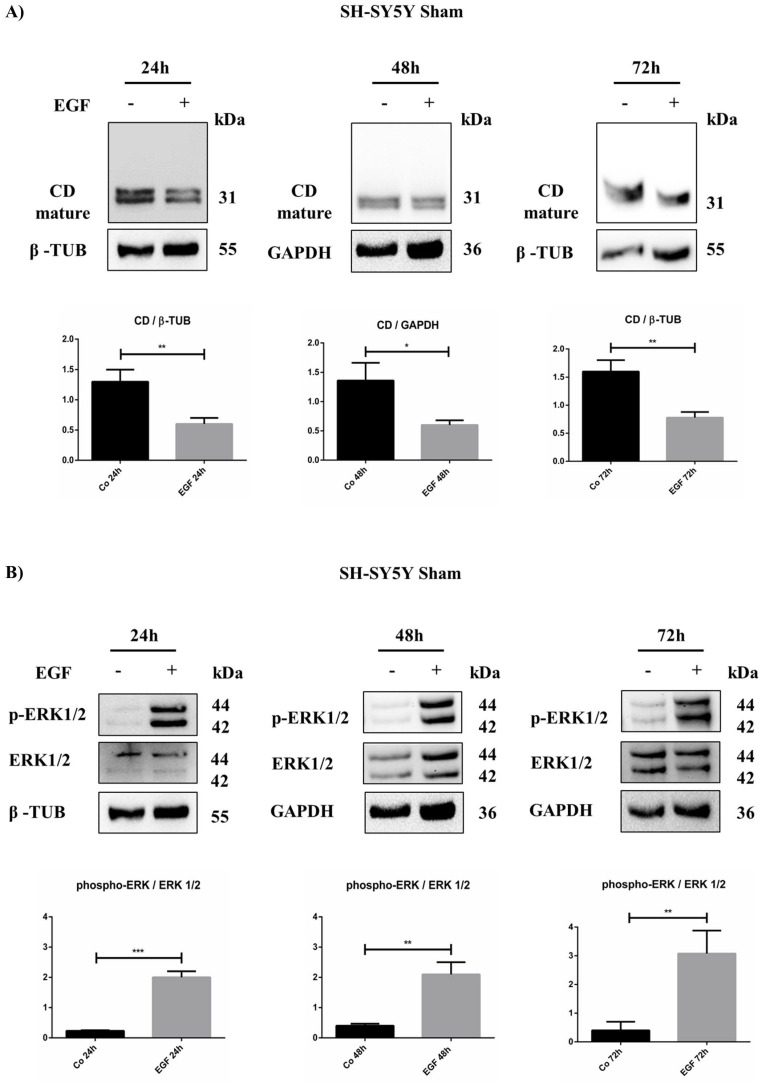
EGF downregulates intracellular cathepsin D in SH-SY5Y Sham. SH-SY5Y cells were treated for 24, 48, and 72 h. (**A**) Time course of cathepsin D expression analyzed through western blotting. (**B**) Western blot of phospho-ERK 1/2 and total protein in Sham cell homogenates at different time points (24, 48, and 72 h) in the presence/absence of EGF. The filters were probed with β-tubulin and GAPDH as loading control. Significance was considered as follows: *** *p* < 0.001; ** *p* < 0.01; * *p* < 0.05.

**Table 1 ijms-23-04782-t001:** Oligonucleotide sequences of both top and bottom strands of shRNA 879, 486 and Ohri are shown. Nucleotides for the directional cloning of the double strand oligonucleotides into the pENTR™/H1/TO vector are shown in italics.

shRNA	Oligo Strand	Sequence
879	Top	5′-*cacc*GCACAGACTCCAAGTATTACACGAATGTAATACTTGGAGTCTGTGC
	Bottom	5′-*aaaa*GCACAGACTCCAAGTATTACATTCGTGTAATACTTGGAGTCTGTGC
486	Top	5′-*cacc*GGATCCACCACAAGTACAACACGAATGTTGTACTTGTGGTGGATCC
	Bottom	5′-*aaaa*GGATCCACCACAAGTACAACATTCGTGTTGTACTTGTGGTGGATCC
Ohri	Top	5′-*cacc*GGCAAAGGCTACAAGCTGTTTCAAGAGAACAGCTTGTAGCCTTTGCC
	Bottom	5′-*aaaa*GGCAAAGGCTACAAGCTGTTCTCTTGAAACAGCTTGTAGCCTTTGCC

**Table 2 ijms-23-04782-t002:** Doubling time calculated for SH-SY5Y Sham and KD-CD clones.

Clone	Doubling Time
Co Sham	38.6 ± 4.8
EGF Sham	29.9 ± 1.9
Co KD-CD	26.8 ± 1.25
EGF KD-CD	19.05 ± 1.23

## Data Availability

Not applicable.

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
