# Peer review of "High Expression of the Lysosomal Protease Cathepsin D Confers Better Prognosis in Neuroblastoma Patients by Contrasting EGF-Induced Neuroblastoma Cell Growth"

_ijms, 2022, doi:10.3390/ijms23094782_

Round 1

Reviewer 1 Report

This work aimed to understand the mechanisms linking EGFR stimulation and cathepsin D expression with neuroblastoma progression and prognosis. The autos  suggest that chemotherapeutics inhibiting the EGFR pathway along with stimulator of cathepsin D synthesis and activity could benefit neuroblastoma prognosis. . The present work
demonstrates for the first time a novel antiproliferative role of cathepsin D that may be exploited to improve neuroblastoma management and treatment

 The paper is generally well written and structured  .It is necessary that it be reviewed by another expert in the field

Author Response

We thank the reviewer for the kind words and for the positive assessment of our manuscript. 

Reviewer 2 Report

The authors uncover chemotherapeutics inhibiting the EGFR pathway along with stimulator of cathepsin D synthesis and activity to be potentially beneficial for neuroblastoma prognosis.

Points to be addressed:

  1. Did the author check for hazards proportionality, perform a multivariable prognostic model and proceed with correction for potential confounding factors for survival impact?
  2. The underlying message here is that more precision and individualized approaches need to be tested in well designed clinical trials – a challenge, but I would be interested in their perspective of how this might be done.
  3. In the frame of this thinking, s is now well known, tumors grow and evolve through a constant crosstalk with the surrounding microenvironment, and emerging evidence indicates that angiogenesis and immunosuppression frequently occur simultaneously in response to this crosstalk. Accordingly, strategies combining anti-angiogenic therapy and immunotherapy seem to have the potential to tip the balance of the tumor microenvironment and improve treatment response: Epidermal growth factor receptor (EGFR) and its ligand heparin-binding EGF-like growth factor (HB-EGF) sustain endothelial cell proliferation and angiogenesis in solid and haematological tumors, and overexpressed HB-EGF stimulates EGFR expression in an autocrine loop (please expand the introduction/discussion section referring to PMID: 31936715).
  4. for all blot figures, densitometry readings/intensity ratio of each band should be included; the whole blot showing all bands and molecular weight markers should be included in the Supplementary Materials;.

Author Response

We thank the reviewer for the positive assessment of our manuscript and for suggesting insightful improvements. 

We have taken into consideration all the criticisms and suggestions amending the text accordingly (all alterations in the text are reported in red), as detailed in the point-by-point answers to reviewer' comments reported below. 

Points to be addressed:

1. Did the author check for hazards proportionality, perform a multivariable prognostic model and proceed with correction for potential confounding factors for survival impact? Thanks for your suggestion. This is a retrospective study and, unfortunately, dataset lacks several information (including treatment status, for instance), and therefore it was no possible to perform multivariable prognostic model and check hazards proportionality. However, to address the reviewer’s concern we performed the log-rank test to determine the statistical significance in between multiple variables (Fig.2C).

2. The underlying message here is that more precision and individualized approaches need to be tested in well-designed clinical trials – a challenge, but I would be interested in their perspective of how this might be done.

Thanks for your suggestion. This is briefly considered in the Discussion section.

3. In the frame of this thinking, it is now well known, tumors grow and evolve through a constant crosstalk with the surrounding microenvironment, and emerging evidence indicates that angiogenesis and immunosuppression frequently occur simultaneously in response to this crosstalk. Accordingly, strategies combining anti-angiogenic therapy and immunotherapy seem to have the potential to tip the balance of the tumor microenvironment and improve treatment response: Epidermal growth factor receptor (EGFR) and its ligand heparin-binding EGF-like growth factor (HB-EGF) sustain endothelial cell proliferation and angiogenesis in solid and haematological tumors, and overexpressed HB-EGF stimulates EGFR expression in an autocrine loop (please expand the introduction/discussion section referring to PMID: 31936715).

Thanks for this useful suggestion. We have elaborated on this, and referred to the suggested paper, in the Discussion section.

4. for all blot figures, densitometry readings/intensity ratio of each band should be included; the whole blot showing all bands and molecular weight markers should be included in the Supplementary Materials.

We have included the densitometry data for each blot in the Figure. Also, please, see the attached Supplementary Materials figure for raw western blot data.

Round 2

Reviewer 2 Report

The authors have clarified several of the questions I raised in my previous review. Most of the major problems have been addressed by this revision.